# Efficient Catalysts of Ethanol Steam Reforming Based on Perovskite-Fluorite Nanocomposites with Supported Ni: Effect of the Synthesis Methods on the Activity and Stability

Marina Arapova [1], Symbat Naurzkulova [2], Tamara Krieger [1], Vladimir Rogov [1] and Vladislav Sadykov [1,*]

[1] Department of Heterogeneous Catalysis, Federal Research Center, Boreskov Institute of Catalysis, Lavrentiev Ave. 5, 630090 Novosibirsk, Russia

[2] Department of Chemistry and Chemical Technology, M. Kh. Dulaty Taraz Regional University, Tole Bi Str., 60, Taraz 080000, Kazakhstan

* Correspondence: sadykov@catalysis.ru; Tel.: +7-383-330-8763

**Abstract:** Catalysts based on perovskite—fluorite nanocomposites with supported nickel $5\%Ni/[Pr_{0.15}Sm_{0.15}Ce_{0.35}Zr_{0.35}O_2 + LaMn_{0.9}Ru_{0.1}O_3]$ were synthesized by three different methods. Structural and surface features of as-prepared samples were elucidated by $N_2$ adsorption, XRD, HR TEM with EDX; reducibility and reactivity were estimated by $H_2$-TPR, and catalytic properties were studied in ethanol steam reforming in the 500–700 °C temperature range. The best catalytic activity without coke accumulation was demonstrated for the $5\%Ni/[Pr_{0.15}Sm_{0.15}Ce_{0.35}Zr_{0.35}O_2 + LaMn_{0.9}Ru_{0.1}O_3]$ catalyst with nanocomposite support obtained by a simple sequential polymeric preparation method. Highly dispersed particles of metallic nickel strongly fixed on the support after the reaction were shown by the HR-TEM and $H_2$ –TPR data. The catalyst provides stable full conversion of ethanol and hydrogen yield above 60% at 600 °C for at least 6 h.

**Keywords:** nickel; perovskite; fluorite; nanocomposites; ethanol; reforming; hydrogen

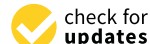

## 1. Introduction

Electrochemical generators based on solid oxide fuel cells (SOFCs) are a part of environmentally friendly, mobile and high-efficient energy technology [1–3]. SOFC with internal and/or external reforming of ethanol as renewable cost-effective fuel has a low exhaust emission and wide application range compared with the other energy production systems [4–6]. Ethanol can be converted to hydrogen via different processes, namely, steam reforming, partial oxidation and oxidative steam reforming [7]. Among them, the steam reforming (SR) is the most widely used one due to the highest hydrogen yield [7,8]. Stoichiometrically, the overall ethanol steam reforming (ESR) reaction could be represented as follows:

$$C_2H_5OH + 3H_2O \rightarrow 2CO_2 + 6H_2 \ (\Delta H\circ = 174 \ kJ/mol) \tag{1}$$

However, ESR consists of many occurring simultaneously reactions, including dehydrogenation (2), steam reforming (3), and the water-gas shift reaction (4) [7].

$$C_2H_5OH \rightarrow CH_3CHO + H_2 \ (\Delta H\circ = 68.9 \ kJ/mol) \tag{2}$$

$$CH_3CHO + H_2O \rightarrow 2CO + 3H_2 \ (\Delta H\circ = 168.8 \ kJ/mol) \tag{3}$$

$$2CO + H_2O \rightarrow 2CO_2 + H_2 \ (\Delta H\circ = 347.4 \ kJ/mol) \tag{4}$$

The ESR process has been extensively studied over noble (Pd, Pt, Rh, Ru) and transition (Ni, Co, Cu) metals-based catalysts. Noble metal-based catalysts, including Pd, Pt,

Rh, Ru exhibit good activity and selectivity in steam reforming of ethanol [9–13]. However, their high costs shifted the attention to inexpensive and highly active transition metals such as Co and, first of all, Ni [7,14]. The main unsolved problem of Ni-based catalysts is the metal sintering and coke formation, which lead to catalyst deactivation and hinder their industrial application. The choice of support plays a crucial role in the regulation of catalyst stability, and complex oxides of rare-earth and transition elements with active oxygen, such as fluorite, perovskite or spinel like complex oxides as catalyst precursor is an attractive approach to minimize the coke deposition on catalysts. Perovskite oxides with $ABO_3$ structure possess high oxygen mobility and, being activated under reducing conditions, show a strong metal-support interaction, therefore suppressing coke formation [15–17]. However, perovskite-type oxides typically have a small specific surface area, therefore sintering of the active metals is inevitable [15,16]. An effective approach to avoid metals sintering and improve the stability of catalysts is to load the active components on a support with a high specific surface area [18–20]. Fluorite-like (doped ceria, zirconia, ceria-zirconia) oxides are widely studied supports for fuel reforming catalysts due to their high surface area, good thermal stability, and oxygen storage capacity [18,20]. As a catalytic layer for fuel reforming in SOFC doped fluorite [6], composites perovskite-spinel [21] and perovskite-fluorite [16,22–24] are extensively studied. These materials should not only provide the reforming reaction but also must satisfy many requirements: compatibility with anode layers and thermochemical resistance to delamination, cracking, sintering, and carbonization [25]. In the early works of our laboratory, perovskite–fluorite-based nanocomposites with very promising characteristics in fuel reforming were developed [26,27].

The methods of nanocomposite materials synthesis should provide a high chemical uniformity of obtained complex oxides along with their high dispersion [28,29]. A lot of effort has focused on obtaining materials with improved characteristics. Among the synthesis methods, the most promising are coprecipitation, hydrothermal method, solvothermal method, sol–gel–citrate and ester polymer precursors (Pechini), microemulsions, microwave method, sonochemical method, solution combustion, spray pyrolysis reactions and one-pot method [30].

In previous work [31], $Pr_{0.15}Sm_{0.15}Ce_{0.35}Zr_{0.35}O_2$ and $LaMn_{0.45}Ni_{0.45}Ru_{0.1}O_3$ were chosen as nanocomposite components for ESR catalyst precursors. Fluorite-like ceria-zirconia solid solution doped with rare-earth cations was chosen as one with high thermal and chemical stability and optimized oxygen mobility/acid-base surface properties [32]. Mn-containing perovskite-type oxides $LnNi_x(M)_{1-x}O_3$ (M = Mn, Fe, Ru) are well-known precursors for highly active catalysts of reforming reactions [33–35], especially ones where nickel exsolution from the perovskite structure occurs during the reduction pretreatment [36,37]. The best catalytic activity along with the coking stability was demonstrated by [$Pr_{0.15}Sm_{0.15}Ce_{0.35}Zr_{0.35}O_2$ + $LaMn_{0.45}Ni_{0.45}Ru_{0.1}O_3$] (1:1 by mass) nanocomposite prepared by the sequential polymeric method due to optimized interaction between components providing the highest concentration of surface metal centers strongly interacting with support and a high reactivity of surface oxygen species. Despite the promising results obtained, the number of accessible nickel atoms on the surface estimated by CO chemisorption, even for the most active sample, was unsatisfactorily low, <1% of the monolayer, which limits its practical application.

This work aims to modify the synthesis method to optimize the content of nickel on the surface, which determines catalytic activity. In the present study, the nickel deposition (5 mass %) was carried out by wet impregnation of the as-prepared [$Pr_{0.15}Sm_{0.15}Ce_{0.35}Zr_{0.35}O_2$ + $LaMn_{0.9}Ru_{0.1}O_3$] (1:1 by mass) nanocomposites. For $LaMn_{0.9}Ru_{0.1}O_3$ perovskite used in this work segregation of Ru in reaction conditions could not result in its decomposition, thus providing higher oxygen mobility in nanocomposite as compared with the previous case [31], which is important for coking suppression. In this paper, the effects of Ni loading and perovskite–fluorite nanocomposite synthesis methods on the textural, structural, surface, and catalytic properties in ESR of obtained materials are presented.

## 2. Results and Discussion

### 2.1. Textural and Structural Properties

Table 1 lists the samples' designation, chemical composition, temperature of calcination ($T_c$), specific surface area (SSA), and a brief description of the synthesis methods of nanocomposites obtained.

**Table 1.** Sample's designation, composition, synthesis methods and SSA.

| Designation | Calculated Chemical Composition | Synthesis Method | $T_c$, °C | SSA, m²/g |
|---|---|---|---|---|
| NiPSCZ_LMRp | | Polymer → wet impregnation | 700 | 21 |
| NiPSCZ_LMRd | 5% Ni/ [$Pr_{0.15}Sm_{0.15}Ce_{0.35}Zr_{0.35}O_2$ + $LaMn_{0.9}Ru_{0.1}O_3$] 1:1 by mass | Dispersion → wet impregnation | 700 | 20 |
| NiPSCZ_LMRo | | One-pot → wet impregnation | 700 | 43 |

The specific surface area of samples is in the range of 20–43 m²/g which is on average lower than that for analogous ones with nickel-containing perovskite in the composition (43–61 m²/g) [31]. This may be due to the difference in the phase composition of the samples or sintering and blocking of micro- and mesopores on the oxides surface during the wet impregnation procedure followed by calcination [38]. Nevertheless, the obtained values of the SSA are satisfactory for use in catalytic applications.

X-ray diffraction patterns (Figure 1) of two samples NiPSCZ_LMRp and NiPSCZ_LMRd contain peaks of the same well-crystallized phases, namely, rhombohedral ($R_{3c}$) perovskite (PDF [89-8775]) and cubic ($F_{m3m}$) fluorite (PDF [028-0271]) as the main phases. In the case of the NiPSCZ_LMRp sample, all peaks are slightly broadened compared to the NiPSCZ_LMRd one, which indicates a greater disorder or dispersion of the phases in the polymer-obtained sample. The pattern of NiPSCZ_LMRo sample demonstrates weakly crystallized mixture of fluorite cubic phase ($F_{m3m}$) oxides with broad blurred peaks and low intensity broadened peaks of the perovskite phase in the region of the strongest diffraction peak (32.7°) (Figure 1). In all three composite samples, the well-defined diffraction peaks of NiO (PDF [65-2901]) are located at the angles 2θ = 37.2°, 43.2°, respectively. Moreover, weak reflections in the 2θ angle region of 30° and 31.5° corresponding to the tetragonal (PDF [080-2155]) and monoclinic (PDF [081-1314]) admixtures of $ZrO_2$ are observed for all three NiPSCZ_LMRp, NiPSCZ_LMRd, NiPSCZ_LMRo samples.

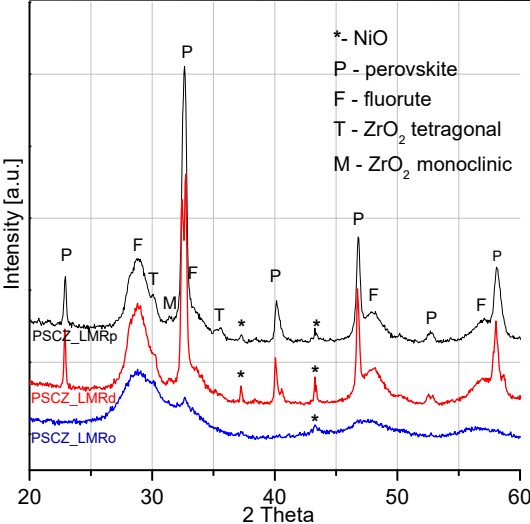

**Figure 1.** The XRD patterns of as-prepared NiPSCZ_LMR samples.

### 2.2. Reducibility

$H_2$ -TPR results are shown in Figure 2a. Three main regions can be distinguished on the curves. At temperatures below 300 °C, there are three overlapping peaks corresponding to the reduction of $Ru^{3+}$ to $Ru^0$, $Ni^{3+}$ to $Ni^{2+}$ and $Mn^{4+}$ to $Mn^{3+}$ [39,40] in perovskite for NiPSCZ_LMRp and NiPSCZ_LMRd samples. The first low-temperature reduction peak at ~180 °C can be attributed to the reduction of active oxygen on the sample surface and well-dispersed NiO weakly bound with support [41,42]. This peak is the most intense for the NiPSCZ_LMRp sample. In the medium temperature range of 300–600 °C, nickel metal particles start to form from the NiO dispersed particles interacting with support. The high –temperature peaks at ~800 °C observed also for separate perovskite phases are usually assigned to reduction of $Mn^{3+}$ to $Mn^{2+}$ state in the bulk of perovskite particles [31]. As can be seen from the XRD results for the samples after reduction in hydrogen at 800 °C (Figure 2b), there is no perovskite structure anymore and reflections of manganese (cubic MnO PDF [05-4310]) and lanthanum (hexagonal $La_2O_3$ (PDF [74-2430]) oxides appear. Also, reflections of metallic nickel (PDF [10-6148]) and ruthenium (PDF [06-0663]) phases are visible for all three samples. Crystalline size estimation from the Scherrer equation for the Ni in reduced samples and NiO in fresh samples gives the same values about 100 nm, which is a limit value for this method. This may be due to the large spread in size for these particles; for a more detailed determination of the surface's particle morphology, TEM after ESR reaction was used. The $H_2$ -TPR profile for NiPSCZ_LMRo looks typical for a mixture of $Mn_xO_y$ and NiO oxides—all reduction peaks are below 500 °C. There are two overlapped peaks with maxima at 237 and 296 °C which can be attributed to the two-step reduction of $MnO_2$: the first step corresponds to the reduction of $MnO_2$ to $Mn_3O_4$ and the second step indicates the further reduction of $Mn_3O_4$ to MnO. This result concurs with the $H_2$ –TPR results for $MnO_2$ reported in the literature [43].

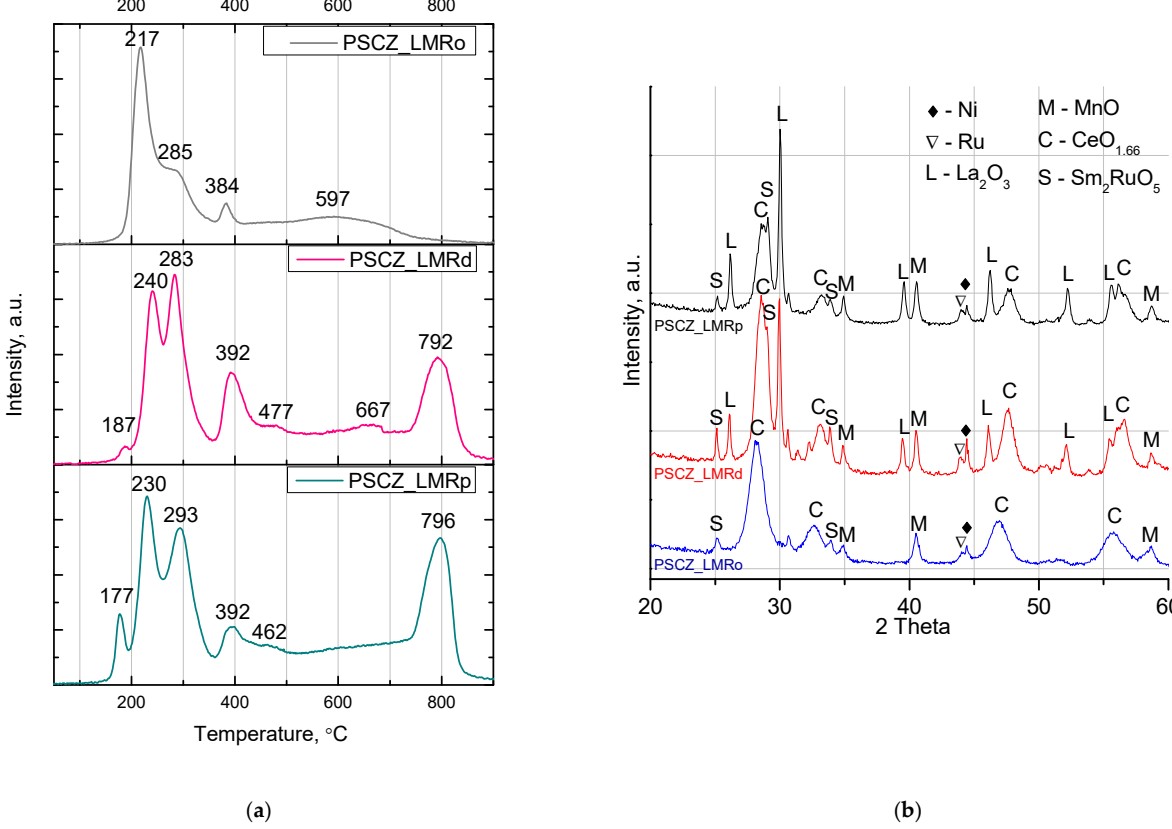

(**a**)  (**b**)

**Figure 2.** (**a**) The $H_2$ -TPR profiles of NiPSCZ_LMR samples; (**b**) The XRD patterns of the samples reduced in $H_2$ at 800 °C.

### 2.3. Catalytic Properties: Ethanol Steam Reforming

All three NiPSCZ_LMR samples showed a high activity in the ethanol steam reforming reaction with the main reaction products being $H_2$, CO, and $CO_2$ in the temperature range of 500–700 °C (Figure 3). Ethanol conversion reaches 100% at 700 °C, and its decrease by decreasing the temperature depends on the oxide support's preparation method. Hydrogen yield at 650 °C for all samples is about 70%, which is close to the thermodynamic maximum and is comparable with the most effective catalysts for the process [7,12,16]. The decrease of the $H_2/CO$ ratio with temperature is explained by the thermodynamic equilibrium of the WGS reaction, which for Ni-loaded doped ceria-zirconia oxides is usually achieved at 350 °C, even at short contact times ~10 ms [44].

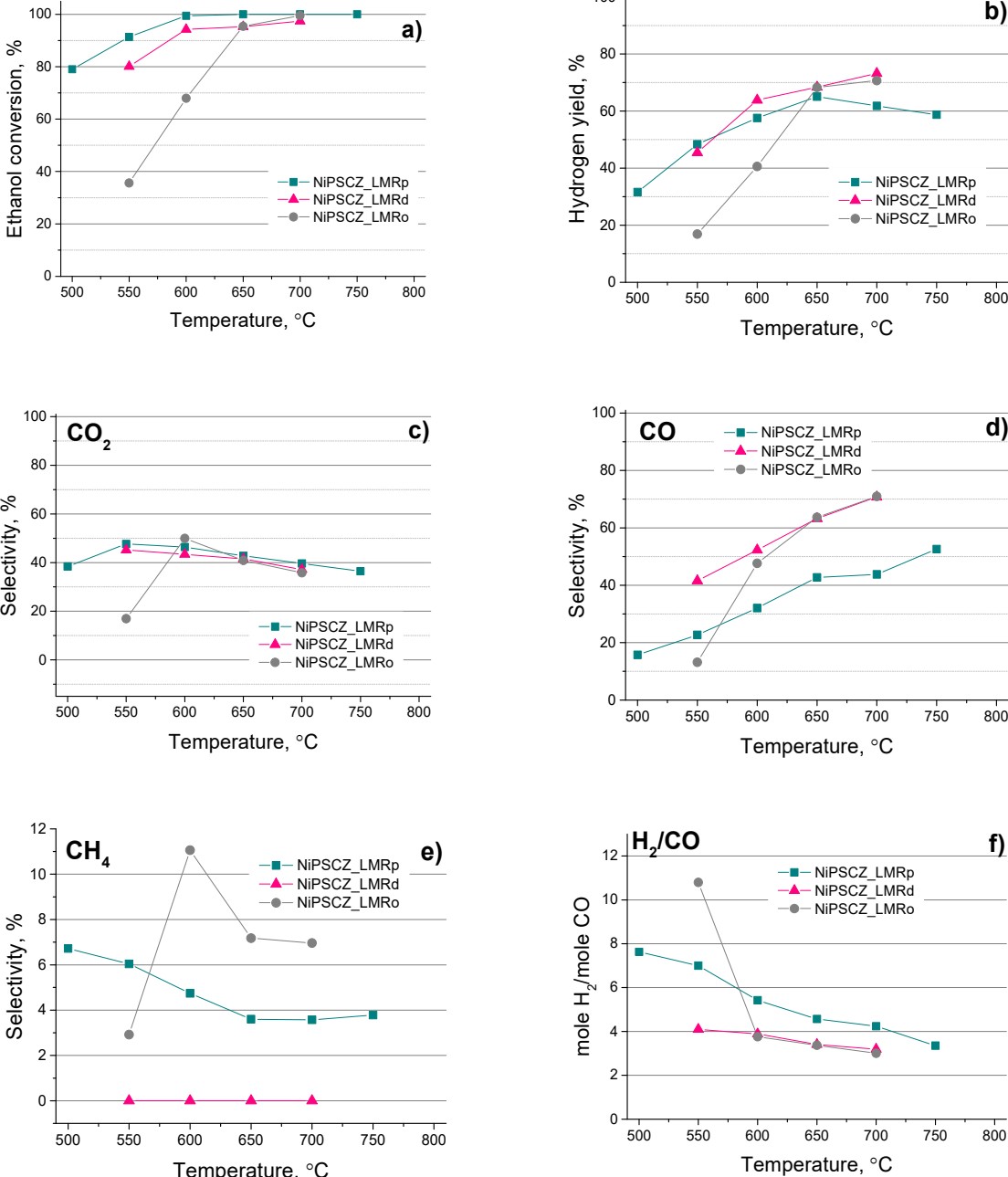

**Figure 3.** Temperature dependencies of (**a**) ethanol conversion; (**b**) hydrogen yield; selectivities of (**c**) $CO_2$; (**d**) CO; (**e**) $CH_4$; and (**f**) $H_2/CO$ ratio for NiPSCZ_LMRp, NiPSCZ_LMRd and NiP-SCZ_LMRo in ethanol steam reforming reaction.

The catalytic behavior is significantly different for samples where perovskite was formed (NiPSCZ_LMRp and NiPSCZ_LMRd) and for one where no perovskite was observed, but only a mixture of oxides (NiPSCZ_LMRo). It can be seen that perovskite-containing samples demonstrate a slight decrease in conversion only starting from a temperature of 600 °C, while for the one-pot sample NiPSCZ_LMRo a significant decrease in activity and hydrogen yield was observed. For temperatures below 600 °C where the decrease in ethanol conversion began the effective reaction rate constant $k_{eff}$ and initial reaction rate $W_0$ were calculated according to Formulas (4) and (5) and are presented in Figure 4 and Table 2.

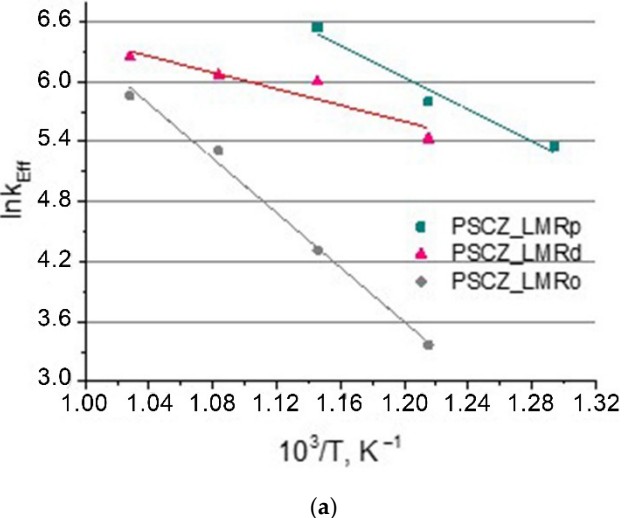

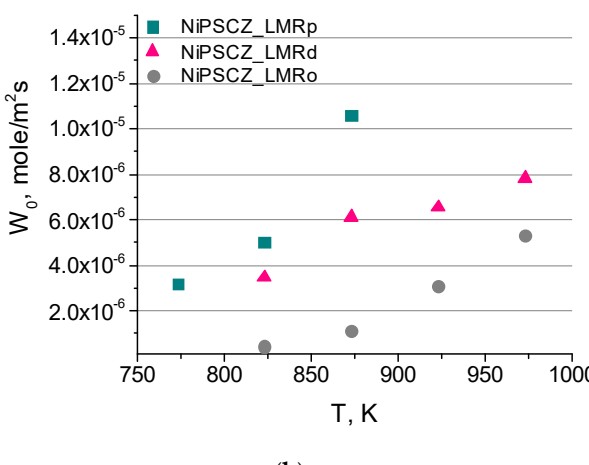

(**a**)  (**b**)

**Figure 4.** (**a**) Linearized Arrhenius equation and (**b**) initial reaction rate $W_0$ for NiPSCZ_LMRp, NiPSCZ_LMRd and NiPSCZ_LMRo in ethanol steam reforming reaction in 500–600 °C temperature range.

**Table 2.** Kinetic regularities calculated for NiPSCZ_LMR and PSCZ_LMNiR * catalysts.

| Title 1 | Ea, kJ/mole | $W_0^{600}$, µmole/m$^2$ s | SSA, m$^2$/g |
|---|---|---|---|
| NiPSCZ_LMRp | 67 | 11 | 21 |
| NiPSCZ_LMRd | 34 | 6.1 | 20 |
| NiPSCZ_LMRo | 113 | 1.1 | 43 |
| PSCZ_LMNiRp * | 61 | 5.4 | 36 |
| PSCZ_LMNiRd * | 96 | 3.3 | 49 |
| PSCZ_LMNiRo * | 109 | 3.1 | 61 |

* More detailed description in [31].

The regularities obtained are similar to those found earlier for samples with nickel introduced at the stage of perovskite synthesis [31] except for NiPSCZ_LMRd one, for which activation energy is only 34 kJ/mole. Since for NiPSCZ_LMRp and NiPSCZ_LMRd samples EtOH conversion in the studied temperature range exceeds 80% (Figure 3), this means that the average composition of the stream in the reactor along the catalytic layer becomes more oxidizing with the temperature and along the catalytic layer due to decrease of the content of highly reactive EtOH molecules. Hence, the state of the Ni active sites is to be more oxidized with the temperature as well, resulting in the decline of their ability to activate ethanol molecules and reaction intermediates. Thus, the estimated activation energy is just an efficient value reflecting such variation in the state of the active sites with temperature. Since the NiPSCZ_LMRd sample middle temperature $H_2$-TPR peaks correspond to the reduction of Ni oxidic species strongly interacting with support dominate (Figure 2), this implies that partial reoxidation by a stream with a low content of ethanol and a high content of steam could be responsible for this phenomenon.

Comparison of the activation energy and the initial reaction rate shows that the highest specific catalytic activity is obtained for the NiPSCZ_LMRp sample, and it correlates well with the less order of the perovskite phase in this composite. As a result, when nickel is deposited on oxide by wet impregnation, its defective structure allows nickel to interact with perovskite more effectively, partially incorporating into its structure, and partially being fixed on the surface. During the reduction of such system, particles of metallic nickel are formed from a number of different states, both by exsolution from the perovskite structure and from nickel oxide of various dispersion strongly fixed on the surface. It is in good agreement with the TPR-$H_2$ data, where a number of high-temperature (above 380 °C) peaks of hydrogen consumption are observed. This ensures strong interaction and high dispersion of active metallic nickel particles, which, in combination with active oxide cations of Mn, Ru, Ce with variable valences provide a bifunctional mechanism for steam reforming reaction.

The time-on-stream catalytic tests at rather low temperature 600 °C for the most active sample NiPSCZ_LMRp demonstrated (Figure 5) that under these conditions the catalyst does not loose activity within 6 h, maintaining the full conversion of ethanol and a constant level of hydrogen yield above 60%. This agrees with apparently higher stability of perovskite particles not containing initially nickel cations, which are reduced in reaction conditions and exsolved as Ni nanoparticles resulting in destruction of the perovskite structure responsible for a high oxygen mobility and reactivity helping to prevent coking.

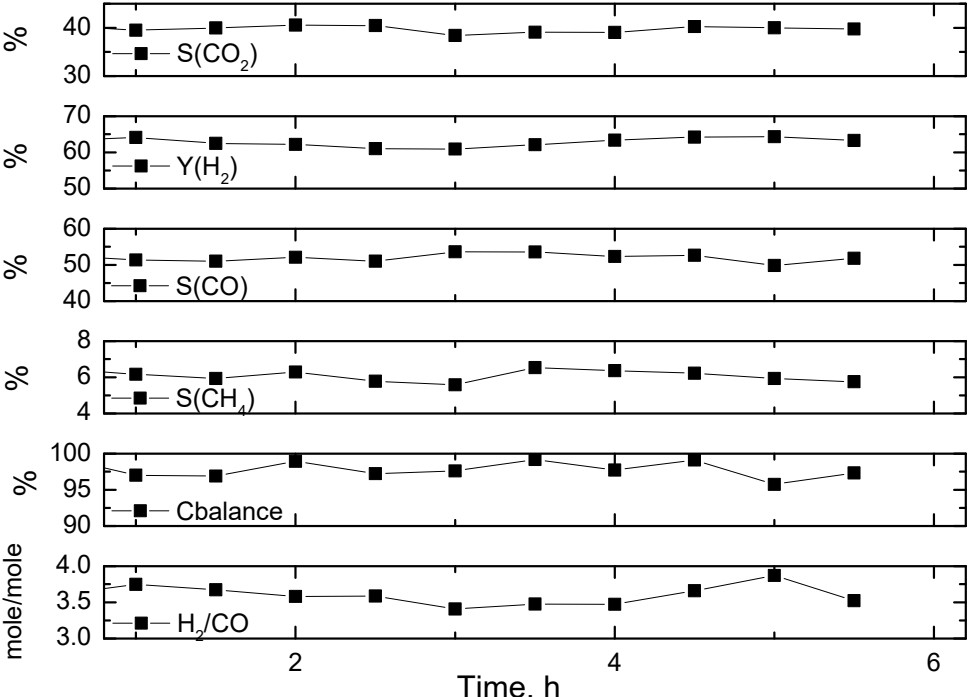

**Figure 5.** Time-on-stream test of NiPSCZ_LMRp catalyst in ethanol steam reforming at 600 °C.

Study of samples after 6 h of reaction by transmission electron microscopy (Figure 6) showed formation of carbonaceous deposits, including a significant amount of carbon filaments, for NiPSCZ_LMRo, while for NiPSCZ_LMRp and NiPSCZ_LMRd only a thin coating of nickel metallic particles with a carbon layer of several nanometers is visible as evidence of an intermediate stage of $CH_x$ formation particles. The absence of carbon deposits for catalysts based on perovskites and composites based on them is practically not observed in the literature, where long-term tests are studied. For example, in operation with a significantly longer contact time of 70 ms, at similar conversion values, there is significant formation of carbon deposits on Ni-containing perovskite-like praseodymium ferrites [15]. The study of the quantitative and qualitative composition of

carbonaceous deposits by TGA and TEM for catalysts after longer experiments (more than 50 h) is planned for further research.

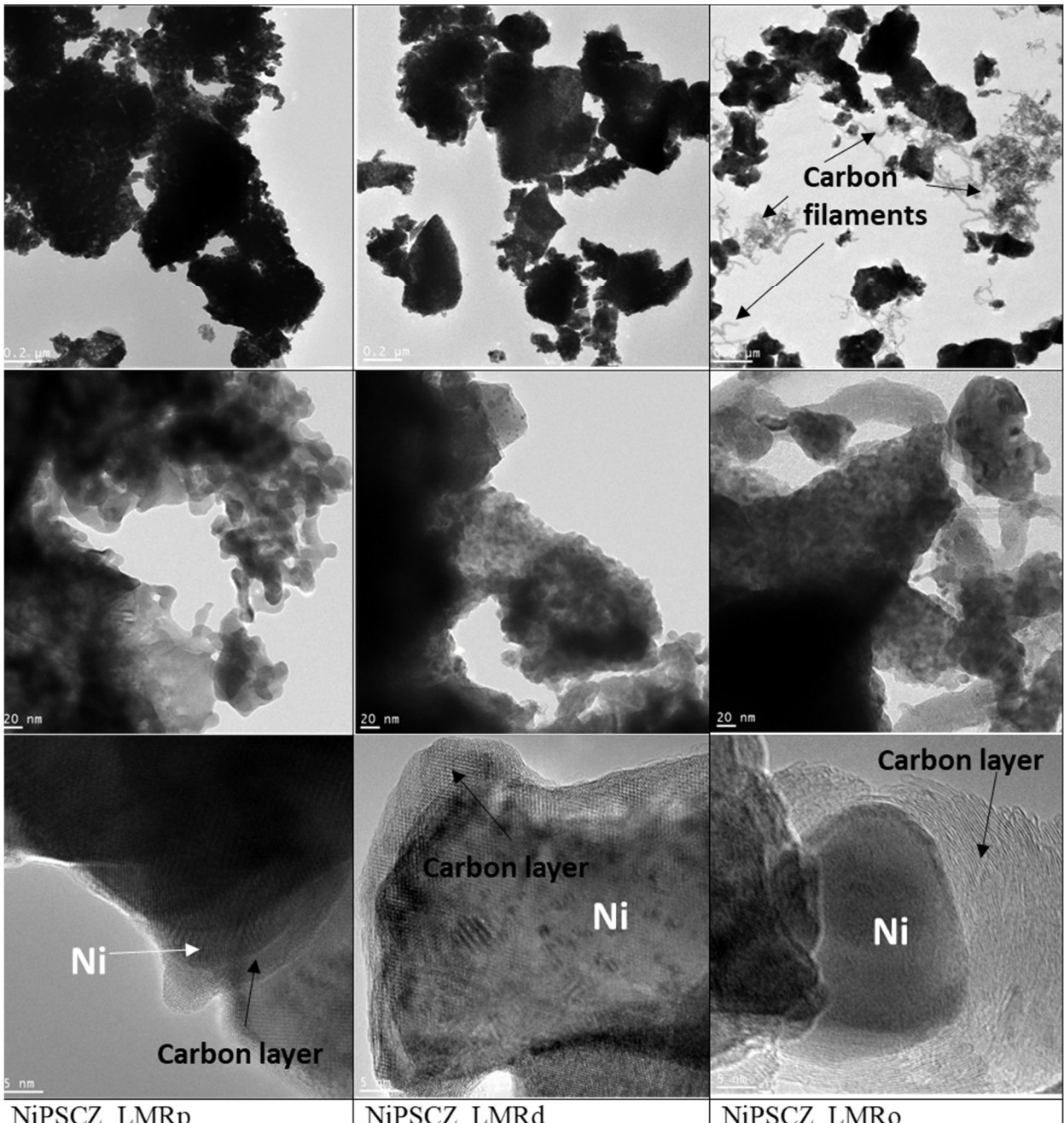

**Figure 6.** The HR TEM images of the catalysts after ESR reaction.

## 3. Materials and Methods

### 3.1. Catalysts Synthesis

$Pr_{0.15}Sm_{0.15}Ce_{0.35}Zr_{0.35}O_2$ (PSCZ) and $LaMn_{0.9}Ru_{0.1}O_3$ (LMR) were synthesized separately from each other by the organic polymeric precursor method (modified Pechini method) [45,46]. All process was carried out under continuous stirring. Citric acid was dissolved in ethylene glycol at 60–80 °C, cooled to a room temperature and supplemented with a necessary number of crystalline nitrates of Pr, Sm, Ce and zirconyl chloride solution for $Pr_{0.15}Sm_{0.15}Ce_{0.35}Zr_{0.35}O_2$ and crystalline nitrates of La, Mn and $RuOCl_3$ for $LaMn_{0.9}Ru_{0.1}O_3$. After salts were completely dissolved, ethylenediamine was added drop-

wise to further polymerize the organic mixture. After complete homogenization during 2 h, the resulting mixture was heated on a hot plate to 200 °C and slowly evaporated to obtain a thick polymer with subsequent decomposition into a solid coal-like mass, which was grinded in a mortar and calcined in air at 700 °C for 4 h.

The reagents molar ratio was: $\nu_{\text{citric acid}}:\nu_{\text{ethylene glycol}}:\nu_{\text{ethylenediamine}}:\Sigma\nu_{\text{metals}}$ = 3.75:11.25:3.75:1. The total moles of metals were calculated according to the formula $\Sigma\nu_{\text{metals}} = \sum_i(i\cdot\nu_{\text{oxide}})$, where i and $\nu_{\text{oxide}}$ are the cation's indices in the chemical formula and the number of moles of synthesized complex oxide.

Nanocomposites $[Pr_{0.15}Sm_{0.15}Ce_{0.35}Zr_{0.35}O_2 + LaMn_{0.9}Ru_{0.1}O_3]$ (1:1 by mass) were prepared by three methods; the sequential polymeric method, ultrasonic dispersion of the as-prepared complex oxides in isopropanol with addition of surfactant and the one-pot polymeric method.

- The sequential polymeric method (sample designated as PSCZ_LMRp): the nanocrystalline fluorite-like oxide $Pr_{0.15}Sm_{0.15}Ce_{0.35}Zr_{0.35}O_2$ was preliminarily prepared by the modified Pechini method in the amount of 2.5 g and calcined in air at 700 °C. After that, a polymer solution containing dissolved metal salts $La(NO_3)_3$, $Mn(NO_3)_2$ and $RuOCl_3$ in a mixture of acetic acid and ethylene glycol was prepared to obtain 2.5 g of perovskite-like oxide $LaMn_{0.9}Ru_{0.1}O_3$. Finally, the calcined fluorite oxide was added to this polymer solution. The resulting suspension was stirred for 2 h at room temperature to achieve homogeneous mixing of the components, after that, it was evaporated at ~80 °C to remove excess water. The obtained thick polymer was ground and calcined in air at 700 °C for 4 h.
- The dispersion method (sample designated as PSCZ_LMRd): both oxides (2.5 g of $Pr_{0.15}Sm_{0.15}Ce_{0.35}Zr_{0.35}O_2$ and 2.5 g of $LaMn_{0.9}Ru_{0.1}O_3$) prepared by the modified Pechini method and calcined in air at 700 °C were added to the solution of 1.5 mL polyvinyl butyral (5%) and 70 mL isopropanol. The resulting mixture was stirred using an IKA T-25 Basic Ultra Turrax Homogenizer for 40 min. After that, the suspension was dried to remove excess solvent and calcined in air at 700 °C for 4 h.
- The one-pot polymeric method (sample designated as PSCZ_LMRo): during one-pot synthesis all cation precursors of both fluorite- and perovskite-like oxides were added simultaneously to the polymer solution of acetic acid and ethylene glycol. The salts of La, Pr, Sm, Ce, La, Mn, and Ru were taken in the corresponding amounts to obtain an oxide mixture $[Pr_{0.15}Sm_{0.15}Ce_{0.35}Zr_{0.35}O_2]:[LaMn_{0.9}Ru_{0.1}O_3]$ = 1:1 by mass. After the complete dissolution of salts, ethylene diamine was added dropwise to further polymerization of the organic matrix. The molar ratio of organic precursors was equal to $\nu$(citric acid):$\nu$(ethylene glycol):$\nu$(ethylene diamine):$\Sigma\nu$(metals) = 3.75:11.25:3.75:1. The resulting solution was evaporated to remove excess of water and obtain a thick polymer, which was ground in a mortar and calcined in air at 700 °C for 4 h.

Nickel (5 wt.%) deposition was carried out by wet impregnation of the as-prepared composites followed by calcination in air at 700 °C for 1 h.

Reagents used (all chemical pure grade): $Pr(NO_3)_3*nH_2O$ (Vecton), $Sm(NO_3)_3*nH_2O$ (Vecton), $Ce(NO_3)_3*nH_2O$ (Vecton), $La(NO_3)_3*n\ H_2O$ (Vecton), $Mn(NO_3)_2*n\ H_2O$ (NevaReactiv), $Ni(NO_3)_2*nH_2O$ (Vecton), $ZrOCl_2$ (Reakhim) and crystalline anhydrous $RuOCl_3$ (Reakhim), $C_6H_8O_7\ H_2O$ (Vecton), ethylene glycol $C_2H_6O_2$ (Reakhim) and ethylene diamine $C_2H_8N_2$ (Reakhim).

For catalytic tests, the calcined oxide powders were pressed into pellets, then crushed and sieved to obtain a 0.25–0.5 mm grain size fraction.

### 3.2. Characterizations

The specific surface area of samples was evaluated by the Brunnauer–Emmet–Teller (BET) method by recording nitrogen physical adsorption at the liquid nitrogen temperature using an ASAP-2400 (Micromeritics Instrument. Corp., Norcross, GA, USA) automated volumetric adsorption unit. Before the analysis, samples were outgassed at 150 °C for 4 h

at a pressure of $1 \times 10^{-3}$ Torr ($\sim$0.1 Pa). The obtained adsorption isotherms were used to calculate the specific surface area.

Diffraction patterns were recorded using a Bruker Advance D8 diffractometer with a CuK$\alpha$ source (2$\theta$ range 20–85°, step size 0.05 and accumulation time 3 s).

High-angle annular dark-field scanning transmission electron microscopy (HAADF-STEM) and high-resolution transmission electron microscopy (HRTEM) images of as-prepared samples were obtained with a JEM-2200FS transmission electron microscope (JEOL Ltd., Japan, acceleration voltage 200 kV, lattice resolution 1 Å) equipped with a Cs-corrector and an EDX spectrometer (JEOL Ltd., Tokyo, Japan). The minimum spot diameter for the step-by-step line or mapping elemental EDX analysis was $\sim$1 nm with a step of about 1.5 nm. Identification of the obtained phases and quantitative calculations were done using the ICDD database.

Material reactivity was characterized by temperature-programmed reduction by $H_2$ (TPR-$H_2$) (10% $H_2$ in Ar, the feed rate 2.5 L/h and the temperature ramp from 25 to 900 °C at 10 °C/min) in a flow kinetic setup with a quartz U-shaped reactor equipped with a Tsvet-500 chromatograph and a thermal conductivity detector.

*3.3. Catalytic Tests*

Catalytic tests in ethanol steam reforming (ESR) were carried out in a continuous flow fixed-bed quartz reactor under atmospheric pressure at a contact time of 10 ms. The catalyst weight of 30 mg (0.25–0.5 mm fraction) was placed between two layers of quartz wool. Preliminary the catalyst was reduced by 5 vol% $H_2/N_2$ (100 mL/min) at 800 °C for 1 h. Ethanol and water were supplied to the general gas flow of He (90 mL/min) by separate saturators with temperatures of 35 and 50 °C, respectively, yielding a feed gas with EtOH:$H_2O$ = 1:4, C(EtOH) = 2% vol. All reagents and products were analyzed by a gas chromatograph Chromos GC1000 (Novosibirsk, Russia,) equipped with TCD and FID detectors.

The tests were performed in two modes, each test was carried out on a separate portion of the sample:

(1) in the temperature range of 500–700 °C. The reaction was started from the highest temperature, at each stage the reaction was carried out until a steady state was reached (about 30 min), two concentration points were recorded, and the temperature was lowered stepwise by 50 °C.

(2) time-on-stream catalytic tests at a constant temperature of 600 °C.

Ethanol conversion X(EtOH), %, hydrogen yield Y($H_2$), %, and products selectivities $S_i$ (i = CO, $CO_2$, $CH_4$, $C_2H_4O$, $C_2H_4$), %, were calculated according to formulas:

$$X(EtOH) = \frac{C(EtOH)}{C^0(EtOH) - C(EtOH)} \times 100 \qquad (5)$$

$$Y(H_2) = \frac{C(H_2)}{6 \times C^0(EtOH)} \times 100 \qquad (6)$$

$$S_i = \frac{\nu_i C_i}{C^0(EtOH) - C(EtOH)} \times 100 \qquad (7)$$

The carbon balance was calculated for all experiments taking into account the concentrations of ethanol (initial and unreacted), CO, $CO_2$, and $CH_4$ using the Formula (8):

$$C_{balance} = \frac{C(CO) + C(CO_2) + C(CH_4)}{C^0(EtOH) - C(EtOH)} \times 100 \qquad (8)$$

For temperatures below 650 °C, the ESR kinetic parameters were calculated with respect to the approximation that the reaction rate is of the first order in ethanol. The

effective reaction rate constant $k_{eff}$, $m^{-2}\,s^{-1}$, was calculated taking into account the specific surface area of samples according to the formulas [30]:

$$k_{eff} = -\frac{\ln(1-X)}{\tau \times SSA \times \rho_{cat}} \qquad (9)$$

where $X$ is ethanol conversion, $\tau$ is the contact time, s, $SSA$ is specific surface area, $m^2 \cdot g^{-1}$, $\rho_{cat}$ is catalyst's apparent density, $g \cdot m^{-3}$. The initial reaction rate was calculated by the formula:

$$W_0 = k_{eff} \times C^0(EtOH) \qquad (10)$$

where $C^0(EtOH)$ is the initial ethanol concentration, M. The effective activation energy was determined by finding the slope of the linearized Arrhenius equation

$$k_{eff} = A \cdot \exp(-Ea_{eff}/RT) \qquad (11)$$

with the form $y = mx + b$; y is $\ln(k_{eff})$, x is $10^3/T$, and m is $-Ea/R$.

## 4. Conclusions

In this study, catalysts precursors based on perovskite–fluorite nanocomposites with supported nickel, i.e., $5\%Ni/[Pr_{0.15}Sm_{0.15}Ce_{0.35}Zr_{0.35}O_2 + LaMn_{0.9}Ru_{0.1}O_3]$ were synthesized by three different methods and tested in ESR. Two synthesis methods—sequential polymeric method and ultrasonic dispersion of as-prepared complex oxides in isopropanol with the addition of surfactant provided the formation of perovskite-fluorite nanocomposites with SSA ~20 $m^2/g$. In the one-pot synthesis method from a polymer containing all cations, the perovskite phase is not formed, and the poor-crystallized mixture of oxides is present. Nickel deposition by wet impregnation followed by reduction leads in the first two cases to the formation of particles of metallic nickel strongly fixed on the complex oxides, which is confirmed by the TPR-$H_2$ and HR-TEM after the reaction data. In the case of the one-pot method, the sample is characterized by large nickel particles weakly bound to the mixture of oxides. The catalytic activity of nanocomposites in ethanol steam reforming is high for all samples in the temperature range of 500–700 °C. The stability of coke formation is determined by the strength of interaction and the particle size of phases where oxidized nickel is stabilized, which, in turn, slightly depends on the method of introducing nickel into a complex oxide precursor and is primarily determined by the structural composition of the oxide support. Nanocomposites NiPSCZ_LMRp and NiPSCZ_LMRd where perovskite formation is observed demonstrate greater resistance to carbonization as a result of the strong interaction of metal particles with the support. Study of these samples after the reaction does not reveal any formation of carbon filaments and bulk particles of coke on the spent catalyst. The most active sample NiPSCZ_LMRp with the highest specific catalytic activity (Ea = 67 kJ/mole, $W_0^{600}$ = 11 μmole/$m^2$ s) provides a stable full conversion of ethanol and hydrogen yield above 60% at 600 °C for at least 6 h.

**Author Contributions:** Conceptualization, M.A., V.S., S.N.; investigation, M.A., S.N., T.K. and V.R.; writing— M.A.; writing—review and editing, V.S.; project administration, M.A. All authors have read and agreed to the published version of the manuscript.

**Funding:** This research was funded by the Ministry of Science and Higher Education of the Russian Federation within the governmental order for Boreskov Institute of Catalysis, project AAAA-A21-121011390007-7.

**Conflicts of Interest:** The authors declare no conflict of interest.

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
