# Peer review of "Efficient Catalysts of Ethanol Steam Reforming Based on Perovskite-Fluorite Nanocomposites with Supported Ni: Effect of the Synthesis Methods on the Activity and Stability"

_catalysts, doi:10.3390/catal12101151_

Round 1
Reviewer 1 Report
This study analyzes the influence the synthesis method of different Ni based perovskite-fluorite nanocomposites has on the ESR. The manuscript is well-structured and the results are of great interest for the potential readers of Catalysts. However, the experimental section must be improved, and a deeper discussion of the results must be included. Moreover, some conclusions are not supported by the results displayed in the manuscript, and therefore further characterization of the synthesized catalysts must be conducted.
My specific comments are presented below:
- In the introduction section, a lot of references are included in line 32. These are too many references for one sentence. Please, revise it.
- A deeper overview of the state of the art concerning the different catalysts used and synthesis methods in the ESR should be included in the introduction.
- The novelty of the work must be emphasized.
- The authors are suggested to include the detail preparation of PSCZ and LMR in this manuscript.
- Ni was incorporated into the as-prepared composites by wet impregnation. Did the authors calcine the catalyst after the impregnation in order to remove the salt precursors? This information is missing in the experimental section.
- How was the catalyst reduction temperature selected? A temperature of 800 ºC is too high and may lead to metal sintering.
- In several parts of the manuscript there is an error in the text: [Error! Bookmark not defined.]: Line 79, line 153, line 265, … Please, revise it.
- In the experimental section, the main reactions involved in the ESR must be included. Moreover, the duration of the reaction should be included.
- How was the mass balance closed? Please, include further details.
- In Table 1, is Tc (calcination temperature) referred to the calcination of the nanocomposites or to the calcination of the catalysts, with Ni being loaded? Please, explain it.
- In lines 78-81, the authors stated that wet impregnation method leads to the decrease of SSA due to the blockage of catalyst pores. The authors must include the SSA of the prepared nanocomposites obtained by different synthesis methods, and prove this statement.
- Figure 1 has some grammatical mistakes: Y-axe: Intensivty, legend: perovkite.
- The XRD profiles of the reduced catalyst would provide more deeper information, since the reduced catalyst form is the one used in the reaction. The authors are suggested to include the XRD analyses of the reduced catalysts at 800 ºC and estimate the Ni crystallite size by Scherrer equation.
- The results displayed in Figure 3, at what time on stream corresponds? In the abstract the authors claimed that the stability of the catalyst is 6 hours, but in the experimental section is not included. Please, carefully revise and clarify this point. The same for the TEM images displayed in Figure 4.
- The spent catalyst samples should be further characterized. The amount of coke deposited in each catalyst must be included and related to the properties of each catalyst prepared by different methods.
- The results obtained in this study must be compared with similar literature results.
- The results obtained in this study must be correlated with the main reactions involved in the process (ESR, WGS….).
- In the conclusion section, the authors claimed that: “The stability to coke formation is determined by the strength of interaction and the particle size of phases where oxidized nickel is stabilized, which, in turn, slightly depends on the method of introducing nickel into a complex oxide precursor, and is primarily determined by the structural composition of the oxide support”. Both the particle size and the oxidation of Ni phases have not been demonstrated by the characterization results. XRD profiles of deactivated catalysts must be included.
- In the conclusion section, the authors claimed that: “Nanocomposites NiP-SCZ_LMRp and NiPSCZ_LMRd where perovskite formation is observed demonstrate a greater resistance to sintering of metal particles and carbonization”. However, metal sintering has not been addressed in this study. The authors must determine the Ni crystallite size of the fresh reduced and deactivated samples by Scherrer equation.
Author Response
Replies attached

Reviewer 2 Report
This paper reports an original investigation of ethanol steam reforming over complex mixed oxides. An active and stable formulation is highlighted. The conclusions of the paper are supported by the data provided. I would suggest only a few minor changes!
1- Define ESR when this abbreviation is first used.
2- Several Bookmarks are missing
3- page 6: the proposed view that the gas stream becomes more oxidizing at higher conversion must be better explained (is it kinetically or thermodynamically?).
4- There are many errors in the numbering of figures (two figure 4, two figure 2.
5- the stoichiometric equation for ESR should be given somewhere.
6- The authors cited CO chemisorption on Ni, while it is well known that the toxic and highly volatile Ni(CO)4 can be formed. This point should be discussed in more details.
Author Response
Replies attached in file, please see

Reviewer 3 Report
The study is dedicated to catalysts based on perovskite (fluorite nanocomposites with supported nickel). The paper presents three different methods for obtaining such materials. The researchers were able to obtain catalysts that provide stable full conversion of ethanol and hydrogen yield above 60% at 600 °C for at least 6 hours.
1. Introduction can be described in more detail and focus on the relevance of the topic and the novelty of the study.
2. A high percentage of the literature used in the preparation of the introduction is from 2004-2014. It is recommended to update the sources in order to show the relevance of the work and the latest achievements in this area.
3. There are technical errors in some parts of the text [6Error! Bookmark not defined]. For example, as in lines 79, 153.
4. Also in the work, you can give examples of comparing the parameters of the obtained samples with typical commercial ones or those described in the literature.
Round 2
Reviewer 1 Report
Following the reviewers advice, the authors have thoroughly revised the paper and clearly responded and clarified their comments and doubts. Since the authors have done a great effort and the paper has been greatly improved I suggest the acceptance of this paper to be publishable.